# 16S rRNA Gene Sequencing-Based Identification and Comparative Analysis of the Fecal Microbiota of Five Syntopic Lizard Species from a Low-Mountain Area in Western Bulgaria

Irina Lazarkevich [1,*], Stephan Engibarov [1], Simona Mitova [1], Emiliya Vacheva [2], Steliyana Popova [3], Nikola Stanchev [3], Rumyana Eneva [1], Yana Gocheva [1], Ivanka Boyadzhieva [1] and Maria Gerginova [1]

1 The Stephan Angeloff Institute of Microbiology, Bulgarian Academy of Sciences, Acad. Georgi Bonchev Str., Bl. 26, 1113 Sofia, Bulgaria; stefan_engibarov@abv.bg (S.E.); simona.mitova9@abv.bg (S.M.); rum_eneva@abv.bg (R.E.); yana2712@gmail.com (Y.G.); petrovaim@abv.bg (I.B.); mariagg@microbio.bas.bg (M.G.)
2 Institute of Biodiversity and Ecosystem Research, Bulgarian Academy of Sciences, 1 Tsar Osvoboditel Blvd., 1000 Sofia, Bulgaria; emilia.vacheva@gmail.com
3 Faculty of Biology, Sofia University "St. Kliment Ohridski", 8 Dragan Tsankov Blvd., 1164 Sofia, Bulgaria; steliyanski@gmail.com (S.P.); nickolastanchev@abv.bg (N.S.)
* Correspondence: irinalazarkevich@abv.bg; Tel.: +359-2-9793139

**Abstract:** Studies on the gut microbiome of free-living reptiles in Europe are generally fragmentary and still missing in Bulgaria. We aimed to identify and compare the fecal microbiota profiles of five syntopic lizard species from three families: the European green lizard (*Lacerta viridis*), the common wall lizard (*Podarcis muralis*), the meadow lizard (*Darevskia praticola*) (Lacertidae), the European snake-eyed skink (*Ablepharus kitaibelii*) (Scincidae), and the European slow worm (*Anguis fragilis*) (Anguidae), which coinhabit a low mountainous area in the western part of the country. A high-throughput sequencing of the hypervariable V3-V4 region of the 16S rRNA gene, performed on the Illumina HiSeq2500 platform, was used. The core microbiota of lizard hosts seems to be species-specific. A dynamic phyla proportion between hosts was found. The richest alpha diversity was observed in *D. praticola*, and the lowest alpha diversity was observed in *P. muralis* and *A. fragilis*. Within the three lacertids, the microbiota of *D. praticola* and *L. viridis* were more closely related to each other than they were to those of *P. muralis*. Sharing a largely common trophic resource (all species except *A. fragilis* are mainly insectivorous) was not an indication of similarity in their gut microbial communities.

**Keywords:** metagenomic analysis; 16S rRNA gene; fecal microbiota; reptiles; lizards; Lacertidae; Scincidae; Anguidae

## 1. Introduction

The development of high-throughput, next-generation sequencing (NGS) technologies and DNA metabarcoding over the past two decades has greatly facilitated the rapid and reliable identification of multiple species from a single sample at the depth required to adequately characterize diverse microbial communities in a given environment/ecological niche, including fastidious and unculturable taxa [1–3]. A metagenomic approach has allowed for a comprehensive identification of the microbiota in different groups of vertebrates [4]. However, most research has focused on mammals [5–9], especially humans [10–14], and less than 10% have been conducted on fish [15–17], amphibians [18–20], reptiles [21–25], and birds [26–28]. Furthermore, the majority of studies are based on fecal samples from captive animals, often from laboratories or zoos [29,30], which tend to exhibit distinct microbiotas compared to their wild counterparts; therefore, the effect of artificial perturbations is well known [30–34]. Host diet and phylogeny are thought to be important predictors of the gut microbiota composition [5,21,35], and host genotype could affect the abundance of some

microbial genera [36]. More investigations need to be conducted to characterize the microbiome in non-model systems to achieve a better understanding of how the environment and host alike shape endogenous microbial communities in wild animal populations [36,37].

Reptiles comprise 17% of all vertebrate species, representing an ancient group with approximately 12,000 extant species, of which ~60% belong to the clade Sauria, known as lizards [38]. To date, investigations on the gut microbiota of some lizard species from different parts of the world have been reported [39–43]. In Europe, although lizards are important components of ecosystems and are among the dominant reptile species in terms of numbers [44], studies are generally scarce [45,46], except for more intensive studies on some *Podarcis* species [47–50]. Five lizard species belonging to three families: the European green lizard (*Lacerta viridis* Laurenti, 1768), the common wall lizard (*Podarcis muralis* Laurenti, 1768), the meadow lizard (*Darevskia praticola* Eversmann, 1834) (Lacertidae), the European snake-eyed skink (*Ablepharus kitaibelii* Bibron and Bory de Saint-Vincent, 1833) (Scincidae), and the European slow worm (*Anguis fragilis* Linnaeus, 1758) (Anguidae). These lizard species coinhabit a low-mountain area in the western part of Bulgaria. They exhibited different preferences in terms of microhabitat selection, food spectrum, and trophic niche width [44]. The aim of this study was to identify and compare the fecal microbiota of those syntopic lizard species by sequencing the hypervariable V3–V4 region of the 16S rRNA gene. So far, research on the herpetofauna with an aspect on the microbiome has not been conducted in Bulgaria. Therefore, this study could illuminate factors shaping the gut microbiota in still understudied reptile hosts.

## 2. Materials and Methods

### 2.1. Study Area and Sample Collection

The study area was located in western Bulgaria, along the valley of the Dalbochitsa River in Ihtimanska Sredna Gora Mountains, northeast of the village of Gabrovitsa (42°15′12″ N 23°53′59″ E), 430–580 m above sea level. Fieldwork was conducted in May and June 2022. A total of 86 specimens were captured: *L. viridis* $n = 15$; *P. muralis* $n = 17$; *D. praticola* $n = 26$; *A. kitaibelii* $n = 26$; and *A. fragilis* $n = 2$. Lizards were caught by hand, and the animals were placed in individual sterile plastic containers and transported to the laboratory for a period of 1–3 days. After defecation, feces were collected using sterile tweezers, transferred in sterile microcentrifuge tubes (Eppendorf® 2 mL), and immediately frozen at −20 °C until they were used for further DNA extraction. After sampling, the lizards were released at their place of capture (the location of each individual was recorded using a GPS device). The handling of animals was performed according to the necessary regulations and ethics requirements.

### 2.2. Genomic DNA Extraction

The total DNA from fecal samples was extracted using HiPurA®Stool DNA Purification Kit (HiMedia, Mumbai, India) according to the manufacturer's protocol. Approximately equal portions of feces from several individuals were transferred into a sterile microcentrifuge tube and mixed to obtain over 250 mg of content per tube. DNA concentration and purity were measured using a NanoDrop 3300 fluorospectrometer (Thermo Fisher Scientific, Waltham, MA, USA) and 1% agarose gel electrophoresis, respectively. Extracted DNA from all probes for a given species was pooled together and processed further as a single sample.

### 2.3. PCR Amplification and Sequencing

The hypervariable V3-V4 region of the 16S rRNA gene was amplified through polymerase chain reaction (PCR) using the universal bacterial primers 341F and 805R. Libraries of the amplicons were generated using a Herculase II Fusion DNA Polymerase Nextera XT Index V2 Kit and sequenced using an Illumina MiSeq platform (2 × 300 bp PE) by Macrogen Inc. (DNA Sequencing Service, Seoul, Republic of Korea).

*2.4. Sequence Assembly and Taxonomic Identification*

The processing of the raw sequencing reads obtained (including overlapping, filtration by sequence quality (Q30), trimming of the primers) was performed using Bioinformatic Software Tools (Illumina, San Diego, CA, USA). The Quantitative Insights Into Microbial Ecology 2 (QIIME2) platform, version 2019.10 [51] was used for the quality control and analysis of data. Assembly of the paired end reads from the original DNA fragments were merged using FLASH (1.2.11). Raw read filtering and trimming to remove the low-quality sequences and chimeras, and operational taxonomic units (OTU) clustering at different distance cutoffs (0.03) were performed using de novo CD-HIT-OUT. Sequences with ≥97% similarity were assigned to the same OTUs using NCBI_16S_20230103 (BLAST). Bacterial taxa with a relative abundance lower than 0.5% were combined into an "Others" class.

*2.5. Bioinformatics and Statistical Analysis*

Alpha diversity (calculated intrasample) and beta diversity (calculated as the dissimilarity between species) were measured using QIIME 2, version 2019.10 [51]. The alpha diversity was based on the number of observed OTUs, community richness (Chao1), diversity (Shannon), evenness (Gini–Simpson), and Good's coverage indexes. To compare the significance of the differences between species, a diversity permutation test based on the Shannon and Simpson indices was performed via PAST 4.07 [52]. Beta diversity was measured using principal coordinate analysis (PCoA) based on both weighted and unweighted UniFrac distance matrices. A vegan package in R software was used for visualization.

## 3. Results

*3.1. Description of the Sequencing Data*

A total of 433 093 read counts (Q30 > 93) ranging from 71,927 to 99,437 across species were obtained. All rarefaction curves asymptotically approached a plateau, suggesting an accurate reflection of the microbial community and indicating a satisfactory level of diversity sampling (Figure 1a). A total of 721 OTUs at the 97% similarity level from all samples were extracted.

*3.2. Diversity of Bacterial Communities*

The alpha diversity of the gut microbiota of *L. viridis*, *P. muralis*, *D. praticola*, *A. kitaibelii*, and *A. fragilis* was characterized by the number of OTUs, Chao1, Shannon's, and Gini–Simpson indices, as summarized in Table 1. Good's coverage index (0.999) indicated that optimally, 16S rRNA gene sequences were extracted from the fecal samples and sufficient sequencing depth was achieved to evaluate bacterial diversity in all five species. The results of the diversity permutation test in terms of the Shannon index, as well as the Simpson index, showed that for all combinations, the differences between species were of a high degree of statistical significance ($p < 0.001$).

The largest number of OTUs and the highest Chao1 and Shannon index values were found in *D. praticola.* The least alpha diversity of the gut bacterial community was observed in *P. muralis* and *A. fragilis.*

Beta diversity in the fecal samples of the five lizard species was compared by referencing unweighted and weighted UniFrac distances. Unweighted UniFrac (qualitative), which is based on presence or absence of observed taxa, is more sensitive to differences in low-abundance features. Weighted UniFrac (quantitave) takes into account the relative abundance of taxa shared between samples, and the impact of low-abundance features is diminished; therefore, it is useful for examining differences in community structure. Both are useful to interpret together. Principal coordinate analysis (PCoA) based on unweighted UniFrac revealed that *L. viridis* and *D. praticola* formed a cluster with the highest level of similarity. *A. fragilis* appeared to be more closely related to the lacertid cluster than *A. kitaibelii*, which represents a separate clade (Figure 1b). In terms of relative abundance, *D. praticola* and *A. kitaibelii* were found to be closest. *P. muralis* was distant from all, not only from the other lacertids (Figure 1c).

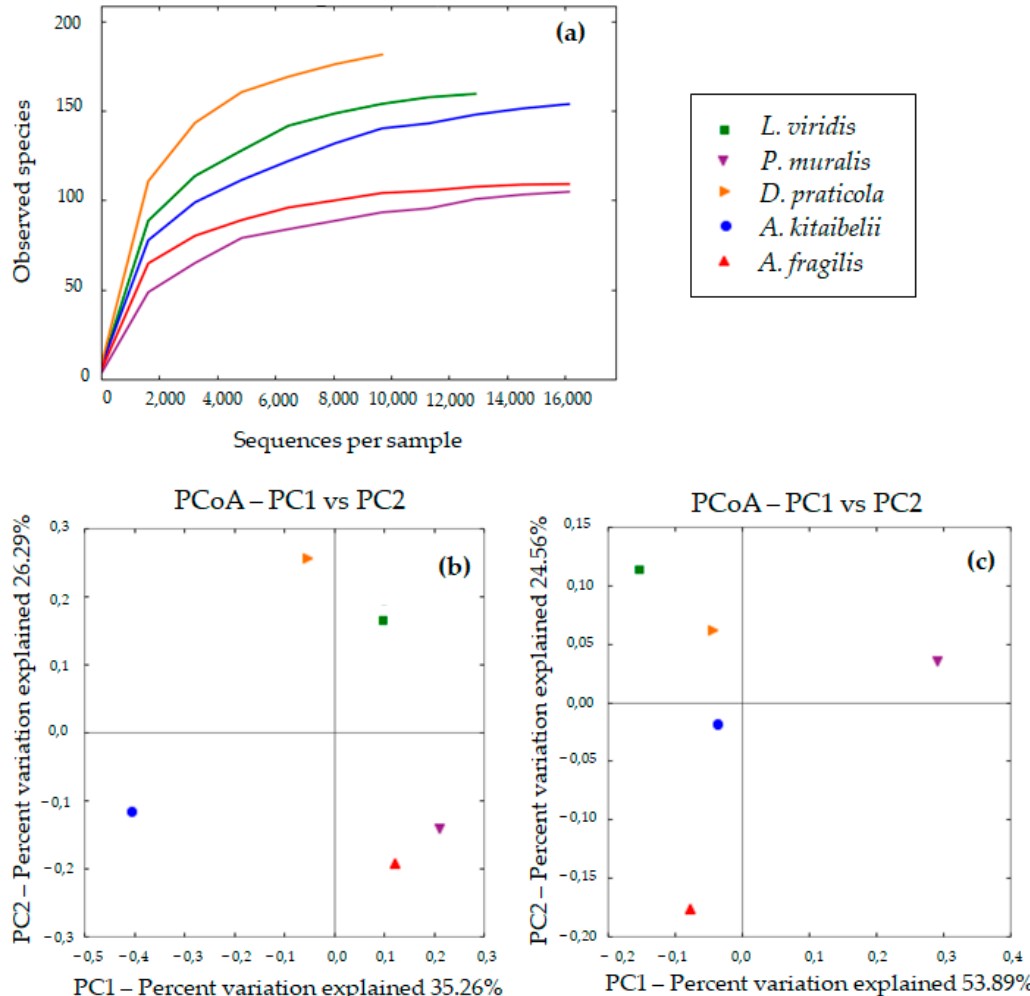

**Figure 1.** Rarefaction curves (**a**) and beta diversity of the gut microbiota of five lizard species by principal coordinate analysis (PCoA): (**b**) unweighted and (**c**) weighted unifrac distance. The variation explained by the plotted principal coordinates is indicated in the axis labels.

**Table 1.** Alpha diversity indexes of the bacterial community in fecal samples of five lizard species.

| Sample | OTUs | Chao1 | Shannon | Gini–Simpson | Good's Coverage |
|--------|------|-------|---------|--------------|-----------------|
| *L. viridis* | 161.0 | 165.71 | 4.54 | 0.886 | 0.999 |
| *P. muralis* | 108.0 | 112.0 | 1.92 | 0.493 | 0.999 |
| *D. praticola* | 185.0 | 202.76 | 5.33 | 0.955 | 0.998 |
| *A. kitaibelii* | 158.0 | 160.65 | 4.52 | 0.919 | 0.999 |
| *A. fragilis* | 109.0 | 111.5 | 3.14 | 0.683 | 0.999 |

### 3.3. Taxonomic Composition and Abundance of Gut Microbiota

A total of six phyla, 12 classes, 21 orders, 43 families, and 106 genera were taxonomically assigned in the overall dataset. The proportion of phyla varied greatly by species (Figure 2a). The most dominant phylum in all lizard species was Bacillota, which ranged between 35.4 and 86.4%, followed by Proteobacteria (0.4–36.8%), Bacteroidota (0.2–23%), Actinomycetota (2.5–18.7%), and Veruccomicrobia (0.4–4.4%), except in *A. fragilis* (55.2%). Cyanobacteria were present only in *L. viridis* (30.8%) (and a negligible incidence < 0.5% in *D. praticola*). Each phylum was at its highest abundance in a different species of lizard: Bacillota in *P. muralis*, Bacteroidota in *D. praticola*, Proteobacteria in *A. kitaibelii*, Verruccomicrobia in *A. fragilis*, and Cyanobacteria in *L. viridis*.

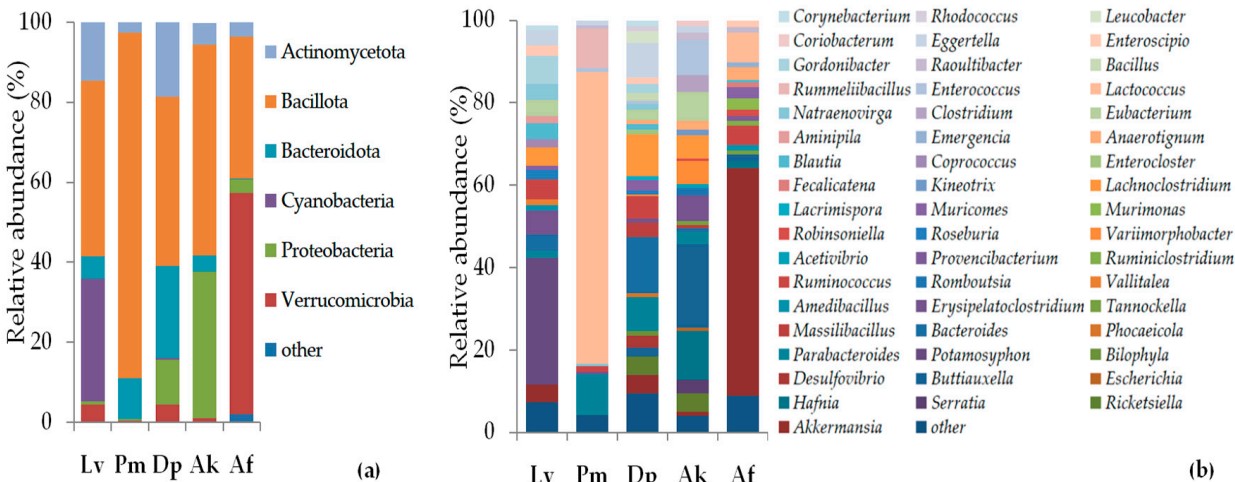

**Figure 2.** Composition of the gut microbiotas of *L. viridis* (Lv), *P. muralis* (Pm), *D. praticola* (Dp), *A. kitaibelii* (Ak), and *A. fragilis* (Af) at (**a**) phylum and (**b**) genus level. One color indicates one taxon in each plot, and the color for "other" combines taxa with relative abundance < 0.5%.

The largest share in the gut microbiota of *L. viridis* was occupied by Bacillota (43.8%), Cyanobacteria (30.8%), and Actinomycetota (14.7%), whereas Bacteroidota (5.6%) and Verrucomicrobia (4.3%) were less represented. A huge portion of the intestinal microbiota of *P. muralis* was occupied by Bacillota (86,4%), followed by Bacteroidota (10.2%) and Actinomycetota (2.5%). Bacillota (42.3%), Bacteroidota (23%), Actinomycetota (18.7%), Proteobacteria (11.3%), and Verrucomicrobia (4.4%) constituted the gut microbial community at *D. praticola*. Bacillota (52.7%) and Proteobacteria (36.8%) were the dominant phyla in *A. kitaibelii*, followed by Actinomycetota (5.4%) and Bacteroidota (4%). In *A. fragilis*, Verrucomicrobia (55.2%) and Bacillota (35.4%) prevailed, whereas Actinomycetota (3.6%), Proteobacteria (3.6%), and unidentified (2%) were less common.

At the class level, overall, the most abundant were Clostridia (ranging between 24 and 34.5%, except in *P. muralis*: 3.9%), Bacteroidia (4–23%, except in *A. fragilis* <0.5%), and Coriobacteriia (2.4–13.4%). Gammaproteobacteria (36.3%) was a prevalent class in *A. kitaibelii*.

At the order level, the most common were Eubacteriales (3.9–34.5%), Bacteroidales (4–23%), Eggerthellales (2.4–13.4%) and Erysipelotrichales (1.2–9.8%).

At the family level, Lachnospiraceae (14.4–20.4%, only in *P. muralis*: 1.6%) and Eggerthellaceae (3.1–13.4%) were represented with the highest abundance overall.

At the genus level, as a whole, *Lachnoclostridium*, *Ruminococcus*, *Eubacterium*, *Erysipelatoclostridium*, *Parabacteroides*, *Bacteroides*, *Eggerthella*, and *Akkermansia* were the most abundant genera. Amongst the prominent genera in *L. viridis* were *Potamosyphon* (30.8%), *Gordonibacter* (6.7%), *Erysipelatoclostridium* (5.9%), *Ruminococcus* (4.9%). In *P. muralis*, *Lactococcus* (70.9%), *Rummeliibacillus* (9.4%), and *Parabacteroides* (9.9%) were the most prominent. In *D. praticola*, *Bacteroides* (13.6%), *Lachnoclostridium* (10.1%), and *Parabacteroides* (8.2%) were found in the most abundance. In *A. kitaibelii,* the most prominent were *Buttiauxella* (20.2%), *Hafnia* (11.9%), and *Enterococcus* (8.6%). In *A. fragilis*, the most abundant were *Akkermansia* (55.2%), *Lactococcus* (7.2%), and *Ruminococcus* (4.6%) (Figure 2b).

The taxa common to the five lizard species encompassed nine classes, nine orders and 12 families. A Venn diagram depicts the number of shared genera between lizard species (Figure 3a). A total of 14 out of 106 recorded genera occurred in all species. The largest number of common genera was between *D. praticola*/*L. viridis* (38) and *D. praticola*/*A. kitaibelii* (33). Differences in terms of the total number of genera, genera with significant presence, and unique genera found in each lizard species are presented in Figure 3b. The underrepresented genera with a relative abundance less than 0.5% occupied between 43.5% and 53.5% of the intestinal microbiota in each species, reaching 71% in *P. muralis*. The

microbial diversity of *A. kitaibelii*, *D. praticola*, and *A. fragilis* featured a high proportion of unique genera (27.8%, 23.9%, and 20%, respectively), followed by *L. viridis* (6.5%) and *P. muralis* (only one genera). However, the participation of unique genera with a relative abundance >0.5% was much lower: *A. kitaibelii* (11.1%), *D. praticola* (8.4%), *A. fragilis* (4.4%), and *L. viridis* (0%). The number of unique species also varied greatly: twenty-two in *D. praticola* (26.5% of the total number of species found in this lizard species), seventeen in *A. kitaibelii* (29.3%), sixteen in *A. fragilis* (34%), seven in *L. viridis* (11.4%), and only one (2.9%) in *P. muralis*.

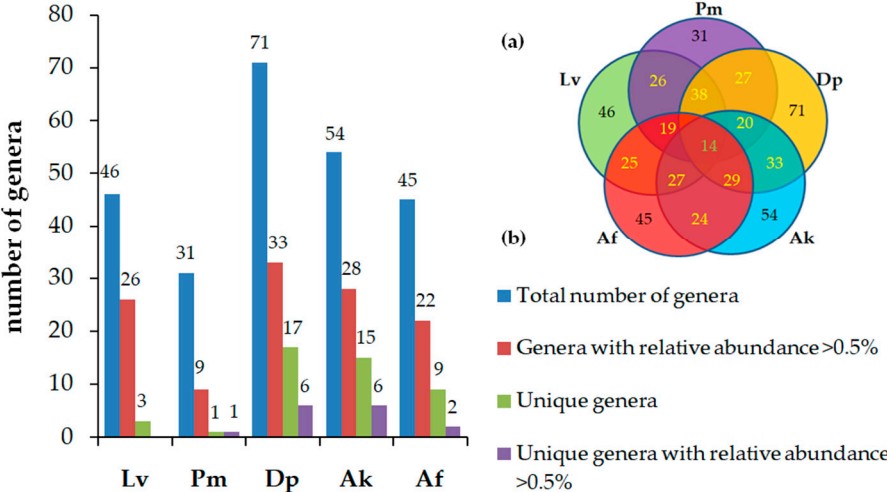

**Figure 3.** Fecal microbiota diversity of *L. viridis* (Lv), *P. muralis* (Pm), *D. praticola* (Dp), *A. kitaibelii* (Ak), and *A. fragilis* (Af) according to (**a**) the number of genera shared between lizard species and (**b**) the number of genera detected in each species.

## 4. Discussion

### 4.1. Composition and Diversity of Lizard Gut Microbiota

Previous reports have indicated that the core gut microbiota of lizards is dominated by the phyla Bacillota (formerly known as Firmicutes) (33.2–73%), Bacteroidota (Bacteroidetes) (6.2–45.7%), and Proteobacteria (5.7–62.3%) [23,31,33,39,40]. We found a dynamic phyla proportion between hosts. Moreover, the core microbiota of each lizard species was dominated by a different phylum. Consistent with other studies, Bacillota (Firmicutes) represented a prevailing phylum in all five lizard species. Firmicutes can encode enzymes related to energy metabolism and the degradation of proteins and other macromolecules [4,31]. Preponderance (except for *P. muralis*) had families Lachnospiraceae (14.4–20.4%), Oscillospiraceae (6–7.2%), and Erysipelothrichaceae (2.2–8.4%), whose members play an essential role in intestinal metabolism. Lachnospiraceae are important producers of butyrate [41], which is involved in maintaining energy homeostasis [53]. *Ruminococcus* sp., a member of Oscillospiraceae, is one of the mutualists adapted to use mucin glycan epitopes, such as fucose or sialic acid, that possesses a unique sialic acid metabolism pathway [54]. Erysipelotrichaceae is probably involved in the host's lipid metabolism [43]. The Bacillota membership in *A. kitaibelii* was distinguished by a higher relative abundance than in other lizard species of *Enterococcus* (8.6%), *Eubacterium* (7.1%), and *Variimorphobacter* (5.6%) and the unique presence of *Clostridium* (4%). Whereas *Eubacterium* (Eubacteriaceae) and *Variimorphobacter* (Lachnospiraceae) produce butyrate, propanol, and acetic acid [55,56], *Enterococcus* (Enterococcaceae) exhibits proteolytic, hydrolytic, and lipolytic activity and produces bacteriocins with antimicrobial properties [57].

Bacteroidota are thought to complement eukaryotic genomes with carbohydrate-processing enzymes (CAZymes) covering a wide range of substrates, in particular polysaccharides, and are responsible for the production of volatile, short-chain fatty acids (SCFAs) (mainly acetate, propionate, and butyrate), which can be reabsorbed by the host, thereby

supporting the total caloric supply [53,58]. This appears to be the phylum with the greatest number of predicted B vitamin producers, as well [59]. Indeed, both Firmicutes and Bacteroidetes contain many different subgroups of bacteria with a variety of properties [21], and the ratio between them (F/B) as the two dominant phyla in gut microbiota has been studied in both humans and animals [60–63]. In general, in reptiles, an increased F/B ratio is associated with hosts having a greater capacity to harvest energy [34]. However, the change in relative abundance of these phyla may be influenced by multiple factors, such as periods of active digestion/fasting [64], free-living/captivity [34], and altitude dependence [65]. F/B ratio could be compared in *P. muralis* and *D. praticola* (in the rest, Bacteroidota was inferior in relative abundance to other phyla). Both species have a similar size and weight, as well as diets close in composition [44]. The high relative abundance of Bacillota in *P. muralis* results in a numerical superiority of the F/B ratio but scarce diversity at the genus level (19 genera vs. 47 in *D. praticola*, 15 shared). Extreme predominance of *Lactococcus formosensis* (70.9%) and *Rumeliibacillus stabekisii* (9.4%), which are both unique to *P. muralis*, probably partially compensates for the lack of diversity. Regarding their possible role in energy balance, *Lactococcus* as a member of lactic acid bacteria, can produce lactic acid and SCFAs with a proven positive effect on the energy metabolism of animals [66], whereas some strains of *R. stabekisii* are potent protease producers [67]. A better-balanced microbial consortium was supposed in *D. praticola*. Therefore, there was more efficient energy acquisition due to the high relative abundance of Lachnospiraceae members *Bacteroides* and *Desulfovibrio*. The latter two genera largely contribute to hydrogen elimination [59]. The removal of hydrogen allows for the more complete oxidation of organic substrates and, therefore, a higher energy yield from anaerobic fermentation [59].

Proteobacteria represents a physiologically and metabolically assorted group of facultative anaerobes relevant for maintaining gut pH and producing carbon dioxide and nutrients for further colonization by strict anaerobes [25]. Its members are able to grow on a range of organic compounds, including proteins, carbohydrates, and lipids, and despite their relatively lower abundance, they contribute to much of the functional variation [68]. It is the dominant phylum in the gastrointestinal tract of some fish [69], snakes [21], and birds [70]. We found the highest relative abundance of Proteobacteria in *D. praticola* (11.3%) and especially, in *A. kitaibelii* (36.8%). Most prevalent in *A. kitaibelii* was Enterobacteriaceae (21.1%), whose members are involved in glucose fermentation and the reduction of nitrates to nitrites [43]. Those with the highest relative abundance were *Buttiauxella warmboldiae* (20.2%), *Hafnia paralvei* (11.9%), and uniquely present *Serratia liquefaciens* (3.3%). *H. alvei* has been reported in various animals, but in reptiles, it has been identified mostly in snakes and skinks [71].

Proteobacteria in *D. praticola* were represented by the Desulfovibrionaceae and Coxiellaceae families. Desulfovibrionaceae have a role in the production of hydrogen sulfide through sulfate reduction, which is an important process for reducing $H_2$ byproducts associated with anaerobic fermentation [39,43]. The main genus of gut sulphate-reducing bacteria is *Desulfovibrio* [59]. It can obtain sulfate from the host via cross-feeding mediated by *Bacteroides*-encoded sulfatase [72]. *D. praticola* harbored a higher relative abundance of both genera (>4%) than the other lizard species (<0.5%). Coxiellaceae belongs to the order Legionellales, which contains several clinically important microorganisms that have been well-studied for their pathogenesis [73]. *Rickettsiella* spp. are obligate intracellular symbionts residing in *Ixodes* ticks and other arthropods, including insects, arachnids, and crustaceans [74]. *Rickettsiella massiliensis* was uniquely present in *D. praticola*.

Verrucomicrobia appeared to be the dominant phylum in the fecal microbiota of *A. fragilis* (55.2%). The only species that has been reported in the gastrointestinal tract— *Akkermansia muciniphila*—relies on mucin as a sole source of carbon, nitrogen, and energy [75]. As a part of specialization to mucin-degradation, it possesses a variety of enzymes for utilizing mucin oligosaccharides, such as proteases, sulfatases, and glycosyl hydrolases, including sialidases [76] and β-galactosidases [77,78]. *A. muciniphila* is a keystone species within the gut microbial community in the mucosal environment because it

increases the availability of mucin sugars and produces acetate and propionate, which serve as substrates for other bacteria and the host [79,80]. The huge proportion of *A. muciniphila* in the gut microbiota of *A. fragilis* probably corresponds to the high mucus content of its diet. *Akkermansia* has also been reported as one of the prevalent genera in the gut microbiota of the mollusk-eating terrapin *Batagur affinis* [81] and in the cold climate-adapted lizards of the genus *Phrynocephalus* [43]. Some studies have indicated that the relative abundance of *Akkermansia* increases under calorie restriction [65]. A mucin-rich environment favors the growth of several other bacteria, except *Akkermansia*, which have the enzymatic capacity to partially or completely degrade mucin, including *Clostridium*, *Lactobacillus*, *Enterococcus*, *Ruminococcus*, and *Bacteroides* [82]. In *A. fragilis*, *L. viridis*, and *D. praticola*, there was higher relative abundance of *Ruminococcus*, as well as members of Lachnospiraceae correlated with the prevalence of *Akkermansia*, which suggests cross-feeding interactions.

Phylum Cyanobacteria was present exclusively in *L. viridis*. A non-photosynthetic group of Cyanobacteria (Melainabacteria) has been regularly identified as a minor (below 1%) component of the microbial populations inhabiting the digestive tract of humans and some animals [9,83–85]. A greater amount of Cyanobacteria has been found to be present in organisms associated with aquatic environments, such as some tadpoles [86]. Cyanobacteria in wild lizards may be related to the ingestion of insects they feed on [31] or those that have a plant-rich diet [39]. Our finding of *Potamosiphon australiensis* (Oscillatoriales) [87] was puzzling because of its high proportion and non-Melainabacteria affiliation. In the specific case of *L. viridis*, however, no environmental or dietary explanation seems plausible. We found no evidence in the available literature of such a widespread intestinal colonization by Cyanobacteria in other reptiles; therefore, this phenomenon remains unclear and demands further attention.

*4.2. Relationship between Gut Microbiotas of Lizard Species and Diet*

In the selected model territory, the syntopic lizard species rely on the same potentially available food resources. The food base and diet of each species have been previously established [44]. We also had this background in mind to compare their fecal microbiotas. The food spectrum of *A. fragilis* is very limited, consisting mainly of snails, insect larvae, earthworms, and centipedes. The remaining four species are insectivorous and, despite their different taxonomic status and body size, largely share a common food base. The trophic niches of the three lacertids and the skink widely overlap, to the greatest extent between *A. kitaibelii* and *D. praticola*, to a lesser extent between *A. kitaibelii* and *P. muralis*, and to the least extent between *L. viridis* and *A. kitaibelii*. According to similarity in food spectrum, *A. kitaibelii* and *D. praticola* formed a cluster and *L. viridis* represented a separate clade [44]. A weighted UniFrac analysis of the gut microbiota accounting for the relative abundance of taxa shared between species also showed the highest relatedness between *D. praticola* and *A. kitaibelii*.

Variety in diet is thought to imply richer microbial diversity, but we did not observe a strict correlation. *D. praticola* had the highest microbial alpha diversity, but it did not have the widest trophic niche. Conversely, *P. muralis* had the least alpha diversity but did not have the narrowest trophic niche. The food niche width was largest in *L. viridis* and narrowest in *A. kitaibelii* [44]. The low alpha diversity of the gut microbiota of *A. fragilis* was expected given the uniformity of its diet.

The hypothesis that dietary similarity is a prerequisite for greater connectivity in gut microbial communities was also not met. Within the three lacertids, *D. praticola* and *P. muralis* exhibited a broader affinity in diet; however, the gut microbiotas of *D. praticola* and *L. viridis* appear to be more closely related. At the phylum level, their relative abundance of Bacillota, Actinomycetota, and Verrucomicrobia was very close (ranging between 42.3 and 43.8%, 14.7 and 18.7%, and 4.3 and 4.4%, respectively) and quite different from the other species. At the genus level, *L. viridis* and *D. praticola* shared 35.8% of genera, whereas only 26.4% was common to the three lacertids.

Commensal and mutualistic bacteria act as a barrier between toxic substances in food and digestive mucosa [37]. Substrates to be degraded by the gut depend largely on the food items that the host consumes [37]. Different microorganisms utilize different substrates, so the available substrates and competition between microbial groups define which microbes will flourish and which will not. Therefore, host diet creates a strong selective pressure on the structure of the gut microbial community [36]. The diversity and functional significance of bacteria, thriving in dependence on food sources, could be indicative of host plasticity. Variations could be driven by the host genotype, specificity in their diet, or a combination of these factors and could be considered a physiological adaptation to dietary heterogeneity. The existence of relevant factors other than diet that shape bacterial diversity at a finer level is suggested.

### 5. Conclusions

Species-specific core microbiota signatures of five syntopic lizard species were identified. The composition and diversity of their intestinal microbial communities differ substantially from phylum to genus levels. *D. praticola* displayed the most diverse gut microbiota. The highest taxonomic similarity was found between *D. praticola* and *L. viridis*. *A. fragilis* exhibited a lowest alpha diversity of the gut microbiota, as well as significant differences in its composition compared to the other studied species due to the specificity of its diet. To elucidate the relationship between the gut microbiota and the factors that shape it, further studies at the individual and population levels are needed. Although the present study is of local importance given the extensive range of each of the lizard species, it sheds light on the still understudied microbial communities associated with animals in the wild environment.

**Supplementary Materials:** The following supporting information can be downloaded at: https://www.mdpi.com/article/10.3390/applmicrobiol4010013/s1, Table S1: Taxonomy abundance count; Table S2: Taxonomy abundance ratio; Table S3: OTU table shared with tax assignment; Table S4: OTU BLAST.

**Author Contributions:** Conceptualization, I.L. and S.E.; methodology, I.B. and M.G.; field work, animal collection and sampling, I.L., E.V., S.P. and N.S.; investigation, I.L., S.E., S.M. and Y.G.; formal analysis, I.L., S.E., S.M., I.B. and M.G.; writing—original draft preparation, I.L.; writing—review and editing, S.E., E.V., R.E. and S.M.; visualization, I.L. and R.E. All authors have read and agreed to the published version of the manuscript.

**Funding:** This research was funded by the National Science Fund of Ministry of Education and Science, Bulgaria (Project KP–06-M51/9, 22 November 2021).

**Institutional Review Board Statement:** Handling of the animals was conducted in accordance with national legislation and permission from the Ministry of Environment and Water (Permit No. 861/13 January 2021).

**Data Availability Statement:** Data are contained within the article or Supplementary Materials.

**Conflicts of Interest:** The authors declare no conflicts of interest. The funders had no role in the design of this study; the collection, analysis, or interpretation of data; the writing of the manuscript; or the decision to publish the results.

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
