# Peer review of "16S rRNA Gene Sequencing-Based Identification and Comparative Analysis of the Fecal Microbiota of Five Syntopic Lizard Species from a Low-Mountain Area in Western Bulgaria"

_2673-8007, doi:10.3390/applmicrobiol4010013_

Round 1

Reviewer 1 Report

Comments and Suggestions for Authors

The study is scientifically relevant and the manuscript is very well written. I just make a few comments.

The microbiota of Anguis fragilis showed significant differences from the other lizard species studied. Could this result not be associated with the low number of animals evaluated?

Authors must include approval of the study by an institutional ethics committee for studies involving animals.

L.68-69: I think it is more appropriate to use 42°15'36.7"N 23°55'14.9"E

L.73: Since Eppendorf is a registered trademark, it is best to refer to the tube as a microcentrifuge tube and include the volume (e.g., 2 mL).

L.188: The data presented in Figure 2B would look better in a table due to the amount of information.

L.223-229, 276-277, 287: Family and phylum names should not be written in italics.

Reviewer 2 Report

Comments and Suggestions for Authors

The author used high-throughput sequencing of the hypervariable V3-V4 region of the 16S rRNA gene, performed on the Illumina HiSeq2500 platform, to identify and compare the fecal microbiota profiles of five syntopic lizard species from three families. In general, the reviewer appreciates the effort that the authors put into the study design. However, there are some weaknesses that need to be addressed:

Major:

In the Discussion section, it is necessary to explain why approximately 80% of the data did not meet the quality standards after data processing (as mentioned in lines 114-115). This raises concerns about the accuracy of the entire sequencing dataset.

It is important to perform statistical analysis on the bacterial abundance of individual lizards. Analyzing only the taxonomic composition may overlook factors such as diet, age, environment, or other variables.

Minor: For abbreviations that appear for the first time, it is recommended to write out the full term first. For example, in line 69, "asl" should be written as "above sea level."

The quality of the images is very poor, with a lot of jagged edges. Please address this issue.

Please explain and discuss the reasons why only two samples of A. fragilis were collected and how this may have affected the results.

Round 2

Reviewer 2 Report

Comments and Suggestions for Authors

Accept in present form